# GATOR1 Mutations Impair PI3 Kinase-Dependent Growth Factor Signaling Regulation of mTORC1

**DOI:** 10.3390/ijms25042068

**Published:** 2024-02-08

**Authors:** Maéline Muller, Jasmine Bélanger, Imane Hadj-Aissa, Conghao Zhang, Chantelle F. Sephton, Paul A. Dutchak

**Affiliations:** Department of Psychiatry and Neuroscience, CERVO Brain Research Centre, Université Laval, Quebec City, QC G1J 2G3, Canada

**Keywords:** GATOR1, NPRL2, NPRL3, mTORC1, PI3 kinase, amino acids, growth factor signaling, metabolism, epilepsy, translation, transcription

## Abstract

GATOR1 (GAP Activity TOward Rag 1) is an evolutionarily conserved GTPase-activating protein complex that controls the activity of mTORC1 (mammalian Target Of Rapamycin Complex 1) in response to amino acid availability in cells. Genetic mutations in the GATOR1 subunits, NPRL2 (nitrogen permease regulator-like 2), NPRL3 (nitrogen permease regulator-like 3), and DEPDC5 (DEP domain containing 5), have been associated with epilepsy in humans; however, the specific effects of these mutations on GATOR1 function and mTORC1 regulation are not well understood. Herein, we report that epilepsy-linked mutations in the NPRL2 subunit of GATOR1, NPRL2-L105P, -T110S, and -D214H, increase basal mTORC1 signal transduction in cells. Notably, we show that NPRL2-L105P is a loss-of-function mutation that disrupts protein interactions with NPRL3 and DEPDC5, impairing GATOR1 complex assembly and resulting in high mTORC1 activity even under conditions of amino acid deprivation. Furthermore, our studies reveal that the GATOR1 complex is necessary for the rapid and robust inhibition of mTORC1 in response to growth factor withdrawal or pharmacological inhibition of phosphatidylinositol-3 kinase (PI3K). In the absence of the GATOR1 complex, cells are refractory to PI3K-dependent inhibition of mTORC1, permitting sustained translation and restricting the nuclear localization of TFEB, a transcription factor regulated by mTORC1. Collectively, our results show that epilepsy-linked mutations in NPRL2 can block GATOR1 complex assembly and restrict the appropriate regulation of mTORC1 by canonical PI3K-dependent growth factor signaling in the presence or absence of amino acids.

## 1. Introduction

GATOR1 is an evolutionarily conserved, intracellular protein complex comprising three requisite subunits called NPRL2, NPRL3, and DEPDC5. Genomic-sequencing studies on people with familial forms of epilepsy [1,2], autism spectrum disorders [3], and individuals who have died of sudden unexpected death in epilepsy (SUDEP) [4] have identified more than 130 mutations across all GATOR1 subunits that are associated with the disease. Deciphering the cellular mechanisms that contribute to GATOR1-dependent disorders has been difficult via clinical observation alone because germline mutations, mosaic mutations, and secondary-hit mutations linked to cortical malformations can occur in affected individuals [1,5]. Recently, however, biochemical studies on the GATOR1 complex have shown the NPRL2 subunit of GATOR1 contains the active catalytic site for its GTPase-activating protein (GAP) function toward RAGA GTP-binding proteins, which activate mTORC1 function when GTP-bound [6]. The impact of disease-linked mutations in NPRL2 on the regulation of mTORC1 by amino acids or other upstream growth factor signaling mechanisms is not clear [7].

GATOR1 functions as an integrative amino acid-signaling complex that mediates cellular homeostasis by regulating the protein kinase called mTORC1 [8,9]. mTORC1 is considered a master kinase that controls cellular growth by regulating translation and anabolic pathways in accordance with nutrient availability and growth factor signal transduction pathways [10,11]. When amino acid levels are sufficient in cells, GATOR1 GAP-activity is blocked by inhibitory protein interactions with a distinct complex called GATOR2, permitting mTORC1 activity and anabolic growth [8]. In contrast, amino acid insufficiency activates GATOR1-dependent RAGA-GTP hydrolysis to inhibit mTORC1 and limit amino acid consumption by downregulating translation, altering transcriptional programs, and stimulating the recycling of metabolites via autophagic processes [12,13]. GATOR1 is further regulated by upstream proteins called Sestrin2 and CASTOR1/2, which are responsible for mediating GATOR1 repression by specific dietary amino acids called leucine and arginine, respectively [14,15]. Importantly, in parallel with GATOR1-dependent amino acid regulation of mTORC1, growth factor receptors that activate PI3K-dependent signal transduction also stimulate mTORC1 function by phosphorylating and repressing tuberous sclerosis complex 1/2 (TSC1/2), the GAP for the lysosomal RHEB GTP-binding protein that activates mTORC1 when bound to GTP [16,17,18]. The dynamic interactions of mTORC1 with RAG and RHEB GTP-binding proteins on the lysosomal surface ultimately control the propagation of signal transduction downstream of mTORC1.

mTORC1 kinase activity stimulates cellular growth processes by regulating the uptake and use of metabolites, in part, via its control of translation and transcriptional programs [10]. When mTORC1 is activated by interactions with RAG-GTP and RHEB-GTP proteins, downstream translational signaling pathways are enhanced by the mTORC1-dependent phosphorylation of ribosomal protein S6 kinase (S6K) at Thr-389 [19], leading to the phosphorylation of ribosomal protein S6 and translation initiation factors [20,21]. When mTORC1 is inhibited, translation is reduced, and transcriptional programs are stimulated, in part, through the phosphorylation-dependent nuclear translocation of transcription factor EB (TFEB) [22,23,24]. TFEB cytoplasmic retention occurs in an mTORC1-dependent manner in NPRL2-defective cells, leading to defects in starvation-induced lysosomal gene expression [25].

Herein, we show the functional effect of epilepsy-linked human mutations in the NPRL2 subunit of GATOR1 on mTORC1-dependent signal transduction. Our study shows that disease-linked mutations can differentially affect mTORC1 function, emphasizing the NPRL2-L105P mutant as a strict loss-of-function protein that prevents GATOR1 assembly and function. Furthermore, we also show that genetic mutations affecting GATOR1 complex assembly restrict the ability to turn off mTORC1 activity when PI3K signaling is blocked by growth factor withdrawal or treatment with the pharmacological inhibitor of PI3K called wortmannin. Collectively, our studies show a novel function of GATOR1 to coordinate the activity of mTORC1 via PI3K-dependent signal transduction in the presence and absence of amino acids.

## 2. Results

### 2.1. Epilepsy-Linked NPRL2 Mutations Differentially Effect mTORC1 Signal Transduction

Missense and early-termination mutations in NPRL2 are associated with GATOR1-dependent neurological disease [1,26]. To determine the functional effect of disease-linked NPRL2 mutants, Arg34* (* = stop), L105P, T110S, and D214H, to regulate mTORC1-dependent signal transduction, we used site-directed mutagenesis to mutate our lentiviral FLAG-NPRL2 expression construct (Appendix A). NPRL2 knockout (NPRL2 KO) mouse embryonic fibroblast (MEF) cells and wild-type (WT) MEF cells were infected with each FLAG-NPRL2 variant to generate stable cell lines. Cells were challenged with starvation in Earle’s buffered saline solution (EBSS) for 1 or 2 h, and the protein extracts were analyzed by Western blotting on mTORC1 kinase targets (Figure 1A). Reintroducing FLAG-NPRL2 into NPRL2 KO MEF cells rescued EBSS-dependent mTORC1 inhibition, permitting the repression of P-S6K (T389) during starvation, like WT cells (Figure 1A). NPRL2-T110S and NPRL2-D214H mutants increased basal mTORC1 signaling but retained the ability to block mTORC1 activity and repress P-S6K during starvation, similar to wild-type cells. In contrast, NPRL2-Arg34* and NPRL2-L105P were unable to rescue GATOR1 function and failed to fully decrease P-S6K during EBSS treatment (Figure 1A). To determine if these mutations had dominant effects, we also analyzed WT MEF cells stably expressing each FLAG-NPRL2 variant. Notably, NPRL2-Arg34* and NPRL2-L105P did not cause dominant negative effects on P-S6K regulation in WT cells (Appendix A). We were unable to detect the expression of FLAG-NPRL2-Arg34* by Western blotting and anticipate that this severely truncated protein is rapidly degraded in cells, limiting our ability to detect the protein. Nevertheless, our observations are consistent with NPRL2-Arg34* being a loss-of-function mutant and support the requirement of the Arg78 residue of NPRL2 being necessary for GATOR1 function [6].

We next mapped the location of each NPRL2 mutation onto the cryo-EM structure (PDB ID 6CET) of the GATOR1 complex [27], including the catalytic site at NPRL2-Arg78 (Figure 1B) [6]. The amino acid residues T110 and L105 reside in the same domain as R78, suggesting that these mutants may alter the dynamic GAP activity of the GATOR1-dependent regulation of mTORC1. To test this, we incubated cells in an EBSS starvation medium for 1 h, followed by the addition of serum-free DMEM for 10 min to inhibit GATOR1 function. The NPRL2-T110S and NPRL2-D214H mutants decreased the phosphorylation of mTORC1 target proteins during starvation and increased their phosphorylation following amino acid treatment, similar to wild-type cells (Figure 1C and Appendix A). In stark contrast, NPRL2-L105P-expressing cells were unable to significantly reduce P-S6K and P-S6 after 1 h of starvation compared with wild-type in a complete medium (Figure 1C and Appendix A). Moreover, stimulating NPRL2-L105P-expressing cells with amino acids after starvation increased P-S6K and P-S6 levels, suggesting that the efficacy of the RAG-dependent GTP exchange factor can be influenced by amino acid abundance in cells. To determine if NPRL2 KO cells expressing NPRL2-L105P retained their sensitivity to pharmacological inhibition of mTORC1, we treated them with either rapamycin or torin1 for 1 h. Consistently, all cells were sensitive to mTORC1 inhibition and reduced the mTORC1-dependent phosphorylation of S6K (Figure 1D). Collectively, these studies suggest NPRL2-L105P is a strict loss-of-function mutant that maintains mTORC1 in a constitutively active state that can be pharmacologically manipulated.

### 2.2. NPRL2-L105P Impairs GATOR1 Complex Formation

Genetic mutations that alter protein-coding sequences can produce a wide variety of functional changes that impact protein–protein interactions or enzymatic properties. We asked if NPRL2-L105P mutant proteins prevent GATOR1 function by altering known protein interactions with NPRL2 or GATOR1. To test this, we generated protein extracts from NPRL2 KO MEF cells expressing either FLAG-NPRL2 or FLAG-NPRL2-L105P and performed size-exclusion chromatography followed by Western blotting of the fractions. Unlike NPRL2, NPRL2-L105P was consistently enriched in low-molecular-mass fractions of <150 kDa (Figure 2A) and unaffected by EBSS treatment (Appendix A). With the caveat of NPRL2 overexpression potentially impacting the biochemistry involved in FPLC, we chose to focus on the distribution of endogenous NPRL3, which we believe is a suitable marker. In this context, our data show a shift in NPRL3 during EBSS treatment, which lacks serum. The Western blot analysis of the NPRL3 subunit revealed less protein in fractions of >66 kDa from cells expressing NPRL2-L105P, suggesting a potential defect in GATOR1 subunit interactions (Figure 2A). We also performed Western blot analysis on the GATOR2 subunit called Sec13 and the target of GATOR1 activity called RAGA, but we did not observe any significant changes in their distribution profiles in either cell line (Figure 2A). These data suggest that NPRL2-L105P is unable to appropriately interact with other GATOR1 subunits.

Next, we used an unbiased approach to determine if NPRL2 or NPRL2-L105P differentially interacts with other cellular proteins. We performed co-immunoprecipitation using NPRL2 KO MEF cells stably expressing the flag-tagged form of these proteins. Co-immunoprecipitated proteins were separated by SDS-PAGE and stained with Coomassie blue to visualize potential interacting proteins (Appendix A). We consistently observed the presence of one clear protein (~160 kDa) that was only present in the NPRL2 immunoprecipitation reaction, but not in NPRL2-L105P or the control. We excised the band from the gel and used mass spectrometry fingerprinting, which identified the unknown target as DEPDC5 with 26% sequence coverage (data not shown). These data suggest that NPRL2-L105P did not appropriately interact with DEPDC5 and prompted us to further test if NPRL2-L105 blocked interaction with all GATOR1 subunits. Again, we used a co-immunoprecipitation method to pull down either FLAG-NPRL2, FLAG-NPRL2-L105P, or NPRL2-T110S, the closest clinical mutation linked to GATOR1 pathologies that retained a biological function. As anticipated, NPRL2 and NPRL2-T110S were able to co-immunoprecipitate with NPRL3 and DEPDC5 proteins, whereas NPRL2-L105P was not (Figure 2B). These data strongly support our unbiased co-immunoprecipitation, mass spectrometry analysis showing that DEDPC5 interacts with NPRL2 but not NPRL2-L105P. Collectively, these mechanistic data strongly support that NPRL2-L105P is a strict loss-of-function mutant because of its inability to integrate into the GATOR1 complex.

### 2.3. GATOR1 Mediates PI3K-Dependent Regulation of mTORC1

Since loss-of-function mutations in GATOR1 are associated with increased mTORC1 signal transduction, we asked if the pharmacological inhibition of PI3K-dependent growth factor signaling could represent an alternative strategy to regulate mTORC1 function and overcome the loss-of-function caused by NPRL2 defects. We treated NPRL2 KO and NPRL2 KO cells stably expressing FLAG-NPRL2 or FLAG-NPRL2-L105P with an inhibitor of PI3K called wortmannin in a complete medium. Wortmannin showed similar efficacy in all cell lines and inhibited PI3K-dependent phosphorylation of P-AKT (S473), P-AKT (T308), and P-TSC2 (T1462) (Figure 3A). However, wortmannin treatment strongly decreased the phosphorylation of mTORC1 targets in control cells, whereas NPRL2 KO or NPRL2-L015 cells showed ~three-fold higher P-S6K levels compared with the control, even after 30 min of treatment (Figure 3A). These data are consistent with the requirement for NPRL2 to mediate the rapid and robust inhibition of mTORC1 when canonical growth factor signaling is impaired, even in the presence of amino acids.

To determine if PI3K-dependent inhibition of mTORC1 is dependent on other GATOR1 subunits, we generated CRISPR-mediated NPRL3 KO MEF cell lines (Appendix A). We confirmed NPRL3 KO by Western blotting on NPRL3 (Appendix A) and further challenged the cells with EBSS starvation to test their impaired ability to block mTORC1 signaling during starvation, similar to NPRL2 KO MEF cells [25] (Appendix A). Consistent with NPRL2-defective cells, NPRL3 KO MEF cells also retained their immediate response to wortmannin by repressing the PI3K-dependent growth factor signaling toward mTORC1, decreasing P-AKT (S473), P-AKT (T308), and P-TSC2 (T1462) (Figure 3B). Remarkably, NPRL3 KO cells were also unable to respond appropriately to wortmannin treatment and maintained more phosphorylation of mTORC1 targets, including P-S6K and P-S6, despite the upstream inhibition of the AKT-TSC signaling axis (Figure 3B) and presence of amino acids in the media.

We next tested if NPRL2-defective cells could repress mTORC1-dependent signaling when challenged with starvation in EBSS or growth factor withdrawal in the presence or absence of wortmannin. Cells were cultured in various media for 1 h, and wortmannin was added for the final 30 min to further restrict PI3K-dependent signaling. Consistent with our previous observations, control cells repressed both upstream and downstream mTORC1-dependent signaling following EBSS treatment, whereas NPRL2 KO and FLAG-NPRL2-L105P cells did not decrease P-S6K to the same extent in EBSS or serum-free media, with or without wortmannin (Figure 3C). We also observed that wortmannin further decreased P-AKT in all cells cultured in EBSS or serum-free media alone, indicating that the drug was active and able to further restrict PI3K activity. However, despite the strong decrease in P-AKT in the absence of growth factors and the presence of wortmannin, NPRL2-defective cells showed elevated levels of mTORC1-dependent phosphorylation of P-S6K compared with control cells (Figure 3C). Collectively, these data show that GATOR1 is necessary to mediate the rapid and robust response of mTORC1 signal transduction during starvation or PI3K inhibition in the presence or absence of amino acids.

### 2.4. Loss of NPRL2 Uncouples PI3K-Dependent Regulation of mTORC1 Functions

PI3K-dependent signaling regulation of mTORC1 has been associated with diverse cellular biological processes, including the regulation of translation and gene transcription. To determine the functional impact of NPRL2 on translational regulation, we performed L-azidohomoalaine (AHA)-labeling of nascent protein synthesis using WT and NPRL2 KO MEF cells treated with wortmannin or cycloheximide, a chemical inhibitor of translation elongation. The incorporation of AHA into nascent translated proteins was detected using Western blotting. Treatment of WT cells with cycloheximide strongly inhibited AHA incorporation into nascent proteins, independent of NPRL2 expression (Figure 4A). In contrast, wortmannin treatment strongly repressed nascent protein synthesis in control cells, but the effect was severely blunted in NPRL2 KO cells (Figure 4A). To verify that wortmannin treatment was effective in the cells, Western blots were re-probed with antibodies against P-AKT (S473) and total AKT. In all cells, wortmannin strongly repressed P-AKT, consistent with PI3K inhibition, as previously shown (Figure 3A). We also used immunoprecipitation and Western blot analysis to determine the GTP-binding state of RHEB in control and NPRL2 KO cells following a 1 h treatment with the vehicle or wortmannin and observed a minor reduction in RHEB-GTP in both cell lines at this time point (Appendix A). These data are consistent with GATOR1 and TSC1/2 functioning in an independent and non-redundant manner to tightly coordinate mTORC1 activity.

We have previously shown that genetic ablation of NPRL2 prevents the starvation-dependent nuclear localization of transcription factor EB (TFEB), a target of mTORC1 kinase activity that undergoes nuclear localization when mTORC1 is inactivated [25]. To determine if NPRL2 defects also block TFEB nuclear translocation mediated by upstream PI3K-dependent signaling, we transfected cells with TFEB-GFP and monitored its subcellular distribution by fluorescent microscopy. Cells were treated in complete media with wortmannin over a 1 h period, and TFEB-GFP localization was captured at 15 min intervals. Cells expressing NPRL2 showed strong nuclear localization of TFEB-GFP 15 min after wortmannin treatment, which increased over the 1 h period of our analysis (Figure 4B). Cells lacking NPRL2 or expressing NPRL2-L105P were resistant to wortmannin-stimulated TFEB-GFP nuclear translocation with a ~three-fold greater retention of TFEB-GFP in the cytoplasm 1 h after treatment. These results support the novel and fundamental requirement for GATOR1 to coordinate the PI3K-dependent repression of mTORC1 in the presence of amino acids. Collectively, these data show that GATOR1 is necessary to rapidly and robustly block downstream mTORC1 functions when PI3K-dependent signaling is impaired (Figure 4C).

## 3. Discussion

GATOR1-dependent regulation of mTORC1 represents an evolutionarily conserved metabolic sensory mechanism to control amino acid consumption in accordance with amino acid availability. The expansive number of GATOR1 mutations linked with epilepsy in humans prompted us to investigate the biochemical function of individual NPRL2 mutants and determine their contribution to mTORC1 signal transduction. Our studies showed that disease-linked NPRL2-34* and NPRL2-L105P mutants block GATOR1-dependent mTORC1 regulation. We further showed that NPRL2-L105P impairs GATOR1 complex formation, likely causing a distortion in the alpha-helix structure where the site resides. Intriguingly, these loss-of-function mutants are functionally distinct from NPRL2-T110S and NPRL2-D214H mutations, which enhance mTORC1 activity in basal conditions yet retain the ability to repress mTORC1 when amino acids are limited. These biochemical differences may account for the severity of NPRL2-associated disease in people. Our observations are consistent with alternative imaging studies that have investigated GATOR1 [28] and emphasize the need for future studies to characterize the molecular impact of other subunit mutations in detail.

While mTORC1 activity is extensively regulated by various upstream regulatory pathways, the reason for this diversity has remained unclear [29]. Significant research has described amino acid-dependent effects on the regulation of RHEB activity, linking amino acid availability to the recruitment of TSC2 at lysosomes [30] and affecting the polyubiquitination of RHEB to interact with mTORC1 [31]. Our analysis of RHEB-GTP binding status following PI3K inhibition for 1 h showed a minor trend of decreased RHEB-GTP. The constitutive association of TSC2 with RHEB during PI3K inhibition has been proposed to sterically hinder the full activation of mTORC1 by RHEB-GTP [7], despite the RAG-mediated localization of mTORC1 in the lysosome. Our mutational studies on NPRL2 suggest that GATOR1 could influence the dynamic recruitment mechanisms of other protein complexes, supporting the observations based on amino acid withdrawal only.

Four distinct RAG GTP-binding protein isoforms (RAGA-D) are known to mediate mTORC1 signaling, with recent studies showing qualitative differences in their capacities to mediate signaling downstream of mTORC1 [32]. Since RAG proteins show tissue-specific expression patterns in vivo, we speculate that deficiencies in each RAG regulatory system could contribute to the penetrance of disease in distinct cells and tissues. For instance, recent studies have described a similar mechanism for mTORC1 regulation by folliculin, a regulator of the RAGC GTP-binding protein that converges with insulin signaling and mTORC1 regulation in tumor growth [33]. The similarities of GATOR1 and folliculin functions are consistent with RAG proteins functioning as important mediators of major growth factor signaling pathways that control cellular homeostasis and metabolism.

We have now demonstrated that defects in GATOR1 cause sustained downstream functions of mTORC1 when PI3K signal transduction is blocked. Cells lacking individual GATOR1 subunits, NPRL2 or NPRL3, or expressing the NPRL2-L105P mutant are unable to rapidly and robustly repress mTORC1 signaling during growth factor withdrawal or treatment with the PI3K inhibitor wortmannin, even in the presence of amino acids. Unrestricted translation in GATOR1-defective cells or tissues likely contributes to proteomic differences that drive metabolic change [34,35]. The sustained effect of mTORC1 signaling on translation could lead to profound metabolic deficiencies that broadly impact important cellular processes like synaptic transmission. Since recent studies have shown significant differences in the abundance of amino acids in different anatomical regions of the brain [36], we anticipate that transient fluctuations in specific amino acids may sensitize people with GATOR1 mutations to seizures at the variable foci associated with GATOR1-dependent familial focal epilepsies (FFEVFs) [1,34]. Neurologically, these deficiencies may occur at discrete times when nutrients are limited, which may provide a rationale for the association of GATOR1 with nocturnal seizures [34,37,38], similar to seizures caused by poor glycemic control in diabetes [39]. Other processes downstream of mTORC1 signaling could further exacerbate cellular defects, at least in part due to a deficiency in TFEB-dependent gene expression and lysosomal function [25,40]. While it is not clear what specific downstream feature of mTORC1 signaling contributes to epilepsy in humans, recent studies using GATOR1 mutant mice have shown therapeutic benefits with chronic treatment with the mTORC1 inhibitor drug called rapamycin [41,42]. These studies raise the possibility of managing at least some GATOR1-dependent diseases by repressing mTORC1 function with inhibitory drugs.

Our research has revealed a novel role of GATOR1 in facilitating the rapid and robust repression of mTORC1 during growth factor withdrawal or PI3K inhibition with drugs. These results underscore the limitations of studies focusing exclusively on growth factor pathways regulating mTORC1 without the context of the GATOR1-dependent amino acid signaling pathway. Through our studies of mutant NPRL2 proteins, our results highlight a new perspective for understanding the important convergence of amino acid and growth factor signaling pathways in orchestrating mTORC1 functions.

## 4. Materials and Methods

### 4.1. Cell Culture and Treatments

Cells were cultured in DMEM (Gibco, Grand Island, NY, USA, 11965-092) supplemented with 10% fetal bovine serum (FBS) and maintained at 37 °C in 5% CO_2_. For amino acid starvation, cells were washed once with phosphate-buffered saline (PBS) (pH 7.4) and incubated in Earle’s balanced salt solution (EBSS) medium for the indicated time points at 37 °C in 5% CO_2_. For the amino acid restimulation experiments, the EBSS medium was replaced with serum-free DMEM, and the cells were incubated for 10 min prior to harvesting. For the drug treatments, DMSO (the vehicle control), 100 nM of wortmannin (Sigma-Aldrich, St. Louis, MO, USA), 250 nM of torin1 (Tocris Bioscience, Bristol, UK), or 10 nM of rapamycin (Tocris Bioscience, UK) was added for the time periods indicated in the figure legends.

### 4.2. Site-Directed Mutagenesis

To generate lentiviral vectors expressing the NPRL2 mutants 34*, L105P, T110S, and D214H, QuickChange site-directed mutagenesis was used to mutate our pLVX-FLAG-NPRL2-IRES-zsGreen1 vector. PCR amplification was performed using PfuUltra II HS DNA polymerase (Agilent Technologies, Inc., Santa Clara, CA, USA) and oligonucleotide primers listed in Appendix A. All plasmids were validated by sequencing performed on the SANGER Sequencing platform at the CHU de Québec-Université Laval Research Center.

### 4.3. Lentiviral Production

HEK293T were transfected using FugeneHD transfection reagent (Promega, Madison, WI, USA, E2311) with VGVS and Δ8.9 packaging plasmids and the lentiviral expression plasmid in a 1:1:1 ratio. The media containing the viral particles were collected 48 h post-transfection, filtered through a 0.45 μM filter, snap-frozen in liquid N_2_, and stored at –80 °C until use.

### 4.4. Generation of Stable Cell Lines

NPRL2 KO cells, previously described in [25], were infected with lentivirus-expressing FLAG-NPRL2 or FLAG-NPRL2 mutants: the 34*, L105P, T110S, and D214H mutations. Following 48 h selection with puromycin, GFP-labeled cells were isolated and expanded. NPRL3 KO MEF cells were generated using a pre-designed NPRL3 CRISPR/Cas9 plasmid consisting of an NPRL3 guide RNA (gRNA) sequence (5′-TCGAGTGAGGTACTGGCAG-3′) cloned to the pEF1a-Cas9-FLAG-2A-Puro (LV05) lentivirus plasmid (Sigma-Aldrich). Lentivirus was used to infect MEF cells pretreated for 10 min with 1 µg/mL of polybrene. Cells were selected with puromycin (2 μg/mL) 24 h post-infection, and single clones were isolated and expanded.

### 4.5. Western Blotting

MEF cells were washed with ice-cold PBS and lysed in 25 mM sodium pyrophosphate, 150 mM NaCl, 50 mM NaF, 5 mM EDTA, 5 mM EGTA, 0.5% sodium deoxycholate, 20 mM HEPES, 10 mM β-glycerophosphate, 1% Triton X-100, and 1 mM Na_3_VO_4_, with EDTA-free protease inhibitors (Roche, Penzberg, Germany). The lysates were centrifuged (at 16,260 rcf, for 10 min, at 4 °C), and the soluble fractions were taken and boiled in Laemmli buffer before performing SDS-PAGE. The membranes were blocked in 5% non-fat dried milk and probed overnight with primary antibodies (Appendix A) at 4 °C. The membranes were incubated with LICOR IRDye secondary antibodies for 1 h at room temperature and washed thrice in TBST before being imaged by the LICOR Odyssey imaging system. The signal intensity was analyzed by densitometry using Image Studio Lite software version 5.2.

### 4.6. Fast Protein Liquid Chromatography

MEF cells were rinsed with ice-cold phosphate-buffered saline (PBS) and lysed on ice in CHAPS lysis buffer (0.3% CHAPS, 100 mM NaCl, 2 mM MgCl_2_, 50 mM HEPES (pH 7.4), 2 mM DTT, 1X protease inhibitors, and 1X PhosSTOP). The lysates were rotated at 4 °C for 5 min and then cleared by centrifugation (at 13,000 rcf, for 20 min, at 4 °C). The supernatants were passed through a 0.45 μm filter, and equal protein amounts were applied to a Superdex 200 10/300 GL column (GE Healthcare, Stockholm, Sweden, 28-9909-44) pre-equilibrated with HEPES buffer (100 mM NaCl, 2 mM MgCl_2_, and 50 mM HEPES (pH 7.4)). Samples were eluted from the column in HEPES buffer at a flow rate of 0.5 mL/min and collected in a total of 26 fractions of 0.5 mL. The fractions were precipitated using 20% (*w*/*v*) trichloroacetic acid (TCA) and incubated on ice for 1 h. The precipitates were collected by centrifugation and washed twice with acetone solution (90% acetone and 0.01 M HCl). The protein pellets were air-dried for 5 min, resuspended in 1X Laemmli buffer, and boiled at 95 °C for 5 min. The proteins from fractions 3–26 were then separated on 10% SDS-PAGE for Western blot analysis. For the estimation of molecular weight (MW), the column was equilibrated using a gel filtration marker kit (Sigma-Aldrich, #MWGF1000) containing the following protein standards: blue dextran (2000 kDa), thyroglobulin (669 kDa), apoferritin (440 kDa), β-amylase (220 kDa), alcohol dehydrogenase (150 kDa), albumin (66 kDa), and carbonic anhydrase (29 kDa).

### 4.7. Co-Immunoprecipitation

Protein extracts were isolated using the lysis buffer detailed in the Western blot methods. Aliquots were taken for the input control or incubated with end-over-end rotation with anti-FLAG M2 magnetic beads (Sigma, USA, #M8823) overnight at 4 °C. The beads were washed four times with the lysis buffer, and the immunoprecipitated proteins were eluted, denatured, and boiled (for 5 min at 95 °C) in 1.5X Laemmli buffer for subsequent Western blot analysis.

### 4.8. Rheb-GTP-Binding Assays

Cells were harvested with an ice-cold lysis buffer (50 mM Tris (pH 8.0), 150 mM NaCl, 10 mM MgCl_2_, 1 mM EDTA, and 1% Triton-X100). The lysates were centrifuged at 12,000× *g* at 4 °C for 10 min before the supernatants were collected for RHEB-GTP immunoprecipitation following the manufacturer’s protocol (New East Biosciences, King of Prussia, PA, USA, #26910) with protein-L agarose beads (Santa Cruz Biotechnology, Santa Cruz, CA, USA, #sc-2336). Briefly, protein-L agarose beads were previously blocked with 5% BSA diluted in a lysis buffer and incubated with equalized protein extracts at 4 °C with end-over-end rotation for 4 h. The beads were washed 3 times with the lysis buffer and eluted, denatured, and boiled in 2X Laemmli buffer. Western blotting was performed against total Rheb E1G1R (Cell Signaling Technology, Danvers, MA, USA, #13879).

### 4.9. Mass Spectrometry Protein Identification

MEF cells transduced with FLAG-NPRL2-expressing plasmids were grown in a full medium. Proteins were harvested in a lysis buffer and pre-cleared by centrifugation, and equalized protein amounts were used for overnight immunoprecipitation with anti-FLAG M2 magnetic beads (Sigma, #M8823). The isolated proteins were resolved on SDS-PAGE and stained in Coomassie Blue R-250. The protein bands were extracted from the gel and processed at the SPARC Biocenter in the Hospital for Sick Children (Toronto, ON, Canada). Briefly, the protein bands were reduced with 10 mM dithiothreitol at 60 °C for 1 h, alkylated with 55 mM iodoacetamide at room temperature for 20 min, and then digested with trypsin (Pierce Biotechnology, Rockford, IL, USA) overnight at 37 °C. The resulting proteolytic peptides were subjected to hydrophobic extraction (5% formic acid and 95% acetonitrile), concentrated using a speedvac, and resuspended in 0.1% formic acid. The peptides were then analyzed by nano-HPLC-MS with an EASY-nLC 1200 system coupled to an Orbitrap Fusion Lumos Tribrid mass spectrometer (Thermo Scientific, Waltham, MA, USA). Peak Studio 10.6 and Scaffold software version 5.0 were used to evaluate the protein identification and quantification.

### 4.10. AHA Labeling of Nascent Translation

Cells were cultured in complete DMEM (Gibco, 11965-092) supplemented with 10% fetal bovine serum (FBS), washed with PBS, and switched to DMEM (Gibco, 21013-024) supplemented with Glutamax, L-cystine, and 5% dialyzed FBS. The cells were treated with 100 nM of wortmannin, 0.1 mg/mL of cycloheximide, or the vehicle control for 45 min. An amount of 50 µM was AHA was added during the treatment to replace methionine in the newly formed proteins, according to the manufacturer’s protocol (Thermo Fisher Scientific). The cells were harvested and proteins were extracted using a lysis buffer supplemented with cycloheximide (0.1 mg/mL). Click-IT reactions were performed according to the manufacturer’s protocol (Invitrogen, Waltham, MA, USA, C10276). The protein lysates were then boiled in 1X Laemmli buffer, separated by SDS-PAGE, and probed with IRDye streptavidin in 5% BSA-TBST. The membranes were analyzed using Image Studio Lite software version 5.2.

### 4.11. TFEB Localization

Cells were cultured in a 6-well plate at 40% confluence and were transfected with FugeneHD (Promega, USA). The fluorescent signal for TFEB-GFP was observed using an Axio Vert.A1 Zeiss microscope with an AxioCam. Images were taken of the cells before and after treatment with 100 nM of wortmannin, following the translocation of TFEB-GFP in the same cells over time at 15, 30, 45 min, and 1 h intervals. Regional intensity measurements of the fluorescent signal in the nucleus and cytoplasm were determined using ImageJ software (version 1.53k), with background signal subtraction for all groups. The ratio of the nucleus/cytoplasm GFP intensity was used to represent the subcellular distribution of TFEB-GFP. The values from each group are represented as means ± SD. Statistical analysis was performed using non-parametric ANOVA with the Kruskal–Wallis test for comparisons between groups.

### 4.12. Statistical Analysis

Statistical analysis was performed using a two-tailed Student’s t-test with Microsoft Excel 2016 unless otherwise indicated. A *p*-value of <0.05 was considered significant.

## Figures and Tables

**Figure 1 ijms-25-02068-f001:**
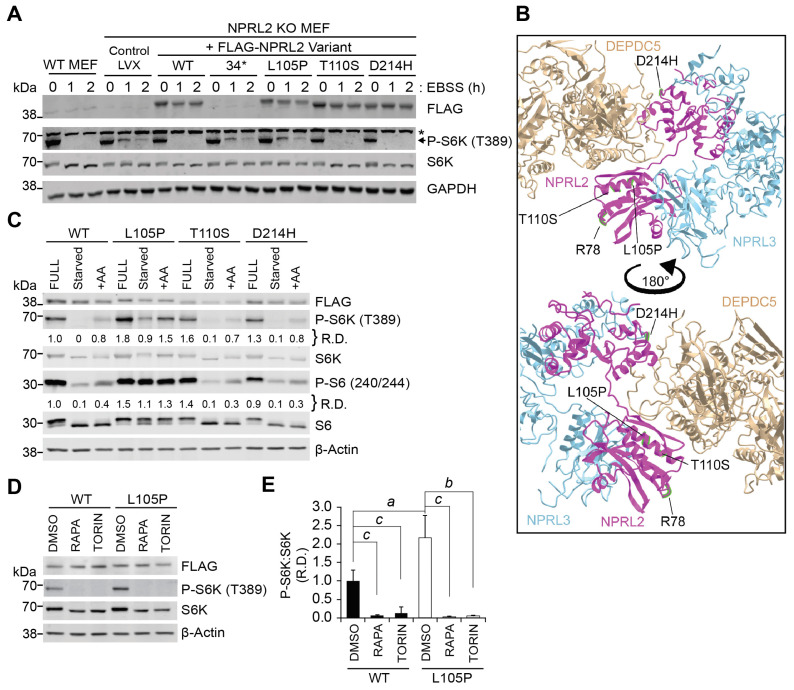
Functional analysis of epilepsy-linked NPRL2 mutations. (**A**) WT and NPRL2 KO MEF cells expressing either FLAG-NPRL2, FLAG-NPRL2-34*, FLAG-NPRL2-L105P, FLAG-NPRL2-T110S, or FLAG-NPRL2-D214H were challenged with EBSS starvation for 1 and 2 h. (* = non-specific protein; LVX = empty lentivirus.) (**B**) Model of GATOR1 indicating the position of target amino acid residues adapted from PDB ID 6CET [27] (purple = NPRL2; blue = NPRL3; gold = DEPDC5). (**C**) NPRL2 KO MEF cells expressing either FLAG-NPRL2 or the indicated NPRL2 mutants were treated with either EBSS for 1 h followed by incubation with serum-free DMEM containing amino acids for 10 min. (**D**) FLAG-NPRL2 and NPRL2-L105P expressing cells were treated with mTOR inhibitors, rapamycin (10 nM), torin1 (250 nM), or vehicle (DMSO), for 1 h. All protein extracts were analyzed by Western blotting. (**E**) Quantification of P-S6K (T389) to S6K signal intensities normalized to β-actin. R.D. = relative density. Data represent *n* = 3 replicate experiments. *a*, *p* < 0.05; *b*, *p* < 0.01; *c*, *p* < 0.001.

**Figure 2 ijms-25-02068-f002:**
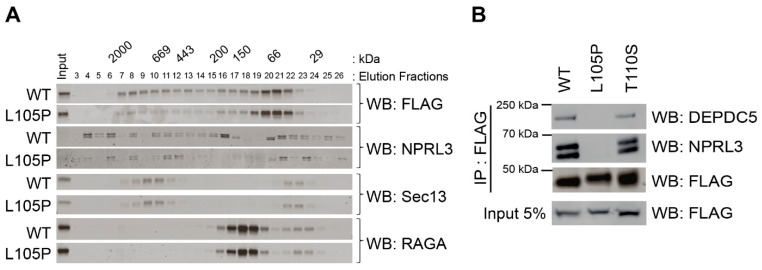
NPRL2-L105P does not interact with GATOR1 subunits. (**A**) Protein extracts from NPRL2 KO MEF cells expressing FLAG-NPRL2 or FLAG-NPRL2-L105P were separated using size-exclusion chromatography. Western blotting of isolated fractions was used to determine the distribution of NPRL2 with the GATOR1 subunit NPRL3, GATOR2 subunit Sec13, and GATOR1 target RAGA. (**B**) Co-immunoprecipitation and Western blot analysis of NPRL2 KO cells expressing FLAG-NPRL2, FLAG-NPRL2-L105P, or FLAG-NPRL2-T110S were used to analyze subunit interactions with NPRL3 and DEPDC5. Western blotting of β-actin shows equal protein input. NPRL2-L105P does not co-immunoprecipitate with NPRL3 or DEPDC5. Data represent 4 independent experiments.

**Figure 3 ijms-25-02068-f003:**
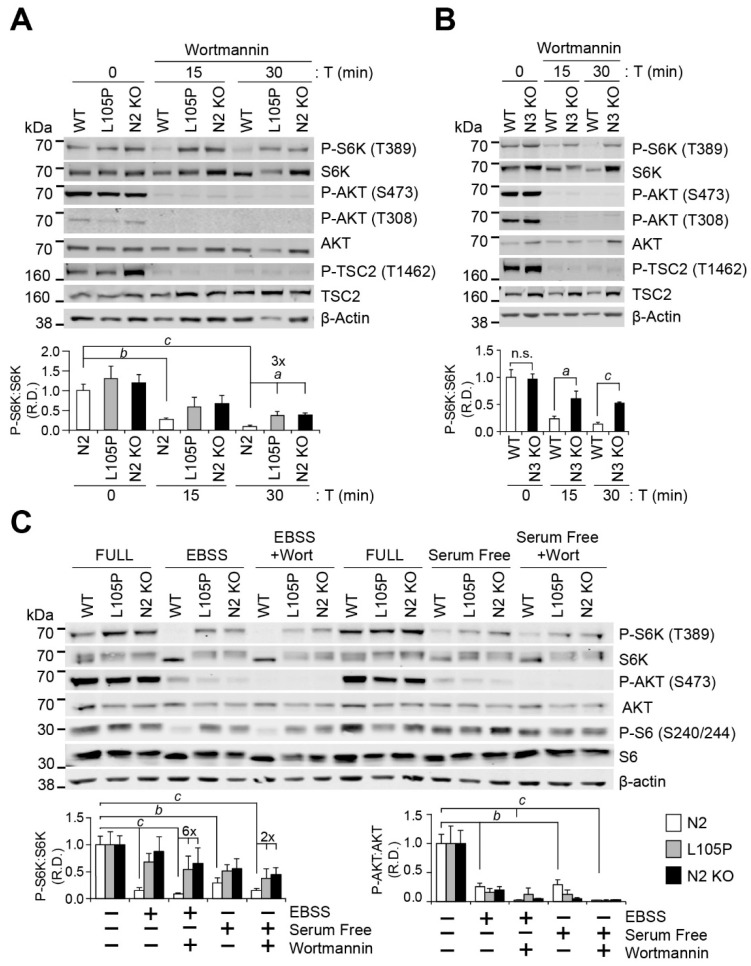
GATOR1 is necessary for rapid and robust repression of mTORC1 via PI3K inhibition. (**A**) WT, NPRL2 KO, and NPRL2 KO cells expressing the NPRL2-L105P mutation or (**B**) WT and NPRL3 KO MEF were cultured in full media and treated with 100 nM of wortmannin for 15 and 30 min; (**C**) WT, NPRL2 KO, and NPRL2 KO cells expressing the L105P mutation were cultured in full media and incubated with either EBSS or serum-free DMEM for 1 h with 100 nM of wortmannin or vehicle (DMSO) treatment. Protein extracts were analyzed by Western blot analysis using the indicated antibodies. Data represent at least 3 independent experiments. *a*, *p* < 0.05; *b*, *p* < 0.01; *c*, *p* < 0.001; n.s. = not significant.

**Figure 4 ijms-25-02068-f004:**
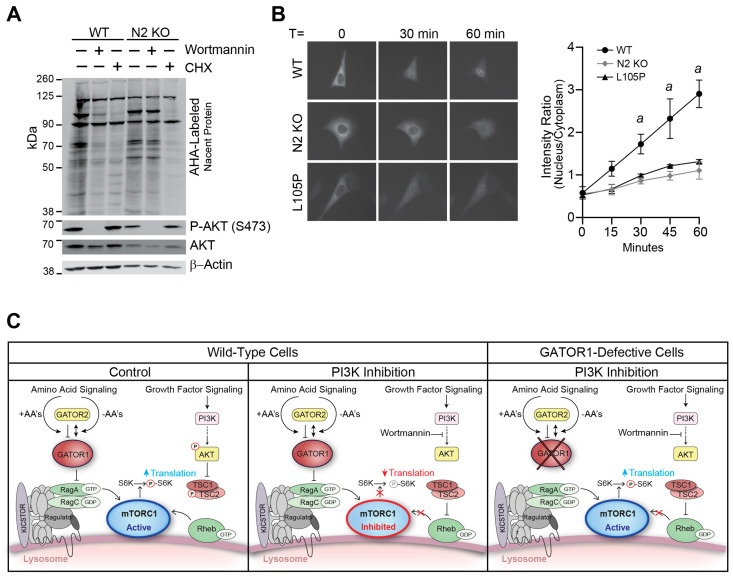
Loss of NPRL2 disrupts PI3K-dependent regulation of mTORC1 functions. (**A**) WT and NPRL2 KO MEF cells were treated with vehicle or 100 nM of wortmannin for 45 min in AHA-labeling media. Proteins were isolated and translated proteins were analyzed using Western blotting. Western blots were also performed on P-AKT (S473), total AKT, and β-actin. (**B**) Impaired nuclear import of GFP-TFEB was captured using epifluorescence microscopy in NPRL2 KO and NPRL2-L105P-expressing cells treated with 100 nM of wortmannin, compared to control cells (at the same magnification). Quantification of GFP-TFEB subcellular distribution (*n* = 3 independent replicates; *a*, *p* < 0.05). (**C**) Model of GATOR1 and growth factor signaling pathways controlling mTORC1 activity. (double headed arrow indicates separation of GATOR1 and GATOR2 during AA-depletion; blunt-line represents inhibitory function).

## Data Availability

Data is contained within the article and Appendix A.

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
