# Peer review of "GATOR1 Mutations Impair PI3 Kinase-Dependent Growth Factor Signaling Regulation of mTORC1"

_ijms, 2024, doi:10.3390/ijms25042068_

Round 1

Reviewer 1 Report

Comments and Suggestions for Authors

In this manuscript entitled " GATOR1 Coordinates PI3 Kinase-Dependent Growth Factor-Signaling 

Regulation of mTORC1”, Muller et al. characterized epilepsy-associated NPRL2 mutations in the mTORC1 signaling pathway and found that GATOR1 complex, which is important for suppressing mTORC1 activity during amino acid starvation, is disrupted by the L105P mutation in NPRL2. In addition, this appears to be the basis for the sustained activation of mTORC1 during amino acid starvation. Regarding the mechanism of mTORC1 activation, the current widely accepted model of mTORC1 activation by amino acids and insulin has two independent axes: the GATOR2-GATOR1-Rag GTPase axis by amino acids and the insulin-induced PI3K-Akt-TSC -Rheb axis by insulin. Interestingly, the authors found that NPRL2 is still required for inhibition of mTORC1 by shutdown of the PI3K-Rheb axis by serum starvation or wortmannin treatment. This is an interesting observation. Indeed, unlike the original mTORC1 activation model, recent studies have shown that the two axes are not independent but interrelated. For example, amino acids can regulate the translocation of TSC2 to lysosomes. Overall, this study provides new insights into mTORC1 regulation by growth factors. However, several points, listed below, need to be improved.

Major points:

In lines 105-106 and lines 296-298, although the authors state that two NPRL2 mutants (T110S and D214H) increased basal mTORC1 activity, this reviewer did not see any increase of basal pS6K1 in Figure 1A and 1B. In addition, the authors did not show the statistical analysis of the elevation in T110S and D214H mutants compared to WT in Figure S1B. Please correct the expression or show the statistical analysis.

At this stage, there is no experimental evidence that growth factor starvation or wortmannin treatment actually causes any changes in GATOR1. In Figure 2A, NPRL2 shows a broad pattern, suggesting that it forms complex associations with other proteins etc. Does growth factor starvation or wortmannin treatment alter the NPRL2 pattern in size-exclusion chromatography? Do these treatments ultimately affect lysosomal localization of mTOR via GATOR1, like amino acid starvation?

In lines 257-259 (Figure S3A), the authors mentioned that Rheb-GTP upon wortmannin treatment was similarly reduced in both WT and NPRL2 KO cells. In my opinion, compared to vehicle, wortmannin treatment should reduce the amount of Rheb-GTP. Howerver, there is no difference in Rheb-GTP levels between the vehicle and wortmannin in Figure S3C. Please check if the reduction of Rheb-GTP level is statistically significant.

In this manuscript, the authors argue that potential crosstalk between Rag axis and Rheb axis to coordinately regulate mTORC1 activity. To discuss these potential crosstalks, it seems better to first mention previous observations on amino acid regulation of the TSC-Rheb axis, and then mention serum-PI3K regulation of GATOR1. The following papers may be helpful for amino acid regulation of the TSC-Rheb axis, and insulin regulation of the Rag axis.

PMID: 24529380; PMID: 26742086; PMID: 33157014; PMID: 34198993; PMID: 37083230

Minor points:

In lines 132-133, “dynamic enzymatic properties of GATOR1”, GATOR1 is a GAP but not enzyme itself. Please modify the expression.

In line 141, typo, P-SK6

Reviewer 2 Report

Comments and Suggestions for Authors

Authors reported that epilepsy-linked mutations in the NPRL2 subunit of GATOR1, NPRL2-L105P, T110S, and -D214H, increase basal mTORC1 signal transduction in cells. This results show that epilepsy-linked mutations in NPRL2 can block GATOR1 complex assembly and restrict the appropriate regulation of mTORC1 by canonical PI3 kinase-dependent growth factor-signaling in the presence or absence of amino acids. The manuscript needs to be improved based on my major comments.

[1]   What is the role of GATOR1 in regulating mTORC1 activity in response to amino acid availability, as described in the research article?

[2]   How do the epilepsy-linked mutations in the NPRL2 subunit (NPRL2-L105P, T110S, and -D214H) affect basal mTORC1 signal transduction in cells?

[3]   What specific molecular mechanisms underlie the loss-of-function phenotype observed in the NPRL2-L105P mutation, and how does it disrupt protein interactions with NPRL3 and DEPDC5?

[4]   In the absence of the GATOR1 complex, how does the cell respond to growth factor withdrawal or PI3 kinase inhibition in terms of mTORC1 regulation?

[5]   Can the disruption of GATOR1 complex assembly by epilepsy-linked mutations in NPRL2 lead to sustained translation in cells, and how does it impact cellular metabolism?

[6]   How does the GATOR1 complex contribute to the inhibition of mTORC1 in response to growth factor withdrawal, and what role does PI3 kinase play in this process?

[7]   What are the downstream effects of sustained mTORC1 activity in cells lacking functional GATOR1 complex, particularly regarding the nuclear localization of TFEB, a transcription factor regulated by mTORC1?

[8]   Are there any potential therapeutic implications for understanding the role of GATOR1 in epilepsy, considering the association of genetic mutations in GATOR1 subunits with the condition?

[9]   How does the research findings contribute to our understanding of the interplay between amino acid availability, growth factor signaling, and PI3 kinase activity in the regulation of mTORC1 by GATOR1?

[10]                       Can the insights from this study be extended to other neurological disorders or conditions beyond epilepsy, where dysregulation of mTORC1 signaling may play a role?

[11]                       Conclusion was missing in the MS. In the conclusion section. The authors should add the significance of this research and its potential practical application.

Comments on the Quality of English Language

Moderate editing of English language required

Reviewer 3 Report

Comments and Suggestions for Authors

The study used NPRL2 KO MEF cells to show that epilepsy-linked mutations in NPRL2 can block GATOR1 complex assembly and restrict the appropriate regulation of mTORC1 by canonical PI3 kinase-dependent growth factor-signaling in the presence or absence of amino acids.

My comments

-The title of the study should be modified to include the type of study and the disease

- Paragraphs from lines 120-148 should be moved before figure 1

- The conclusion should be improved to provide possible preclinical and clinical translation of this study 

- Some typo errors are present, for example, line 348

Comments on the Quality of English Language

Minor editing of the English language is required,  some typo errors are present, for example, line 348.

Round 2

Reviewer 1 Report

Comments and Suggestions for Authors

The authors have addressed most of my previous concerns. However, I recommend showing Figure S2A data along with Figure 2A, to compare the NPRL3 shift upon EBSS treatment (serum+AA starvation). At least, the authors should mention the shift in the text.

“With the caveat of NPRL2 overexpression potentially impacting the biochemistry involved with of FPLC, we chose to focus on the distribution of endogenous NPRL3, which we believe is a suitable marker. In this context, our data do show a shift in NPRL3 during EBSS treatment, which lacks serum.”

Line 314, Typo;  “NPRL2 and NPRL2 suggest”

Author Response

We have now included the experimental caveat in the results section.  I believe this is an important point for new researchers to recognize.  Thank you for the suggestion.

We have also fixed the typo. 

Reviewer 2 Report

Comments and Suggestions for Authors

Requested corrections were completed.

Author Response

Thank you for reviewing our paper.